# Metalearned Neural Memory

**Tsendsuren Munkhdalai, Alessandro Sordoni, Tong Wang, Adam Trischler**
Microsoft Research
Montréal, Québec, Canada
tsendsuren.munkhdalai@microsoft.com

## Abstract

We augment recurrent neural networks with an external memory mechanism that builds upon recent progress in metalearning. We conceptualize this memory as a rapidly adaptable function that we parameterize as a deep neural network. Reading from the neural memory function amounts to pushing an input (the key vector) through the function to produce an output (the value vector). Writing to memory means changing the function; specifically, updating the parameters of the neural network to encode desired information. We leverage training and algorithmic techniques from metalearning to update the neural memory function in one shot. The proposed memory-augmented model achieves strong performance on a variety of learning problems, from supervised question answering to reinforcement learning.

## 1 Introduction

Many information processing tasks require memory, from sequential decision making to structured prediction. As such, a host of past and recent research has focused on augmenting statistical learning algorithms with memory modules that rapidly record task-relevant information [38, 8, 41]. A core desideratum for a memory module is the ability to store information such that it can be recalled from the same cue at later times; this reliability property has been called *self-consistency* [41].

Furthermore, a memory should exhibit some degree of *generalization*, by recalling useful information for cues that have not been encountered before, or by recalling information associated with what was originally stored (the degree of association may depend on downstream tasks). Memory structures should also be *efficient*, by scaling gracefully with the quantity of information stored and by enabling fast read-write operations.

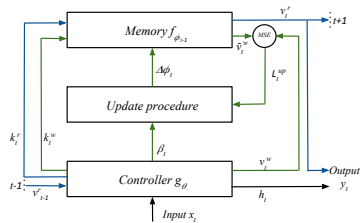

In the context of neural networks, one widely successful memory module is the soft look-up table [8, 46, 3]. This module stores high-dimensional key and value vectors in tabular format and is typically accessed by a controlled attention mechanism [3]. While broadly adopted, the soft look-up table has several shortcomings. Look-up tables are efficient to write, but they may grow without bound if information is stored naïvely in additional slots. Usually, their size is kept fixed and a more efficient writing mechanism is either learnt [8] or implemented heuristically [35].

Figure 1: Schematic illustration of the MNM model. Green and blue arrows indicate data flows for writing and reading operations, respectively. $k_t^r$, $k_t^w$, $v_t^w$ and $\beta_t$ denote read-in key, write-in key, target value and update rate vectors.

The read operation common to most table-augmented models, which is based on soft attention, does not scale well in terms of the number of slots used or in the dimensionality of stored information [32]. Furthermore, soft look-up tables generalize only via convex combinations of stored values. This burdens the controller with estimating useful key and value representations.

In this paper, we seek to unify few-shot metalearning and memory. We introduce an external memory module that we conceptualize as a rapidly adaptable function parameterized as a deep neural network. Reading from this "neural memory" amounts to a forward pass through the network: we push an input (the key vector) through the function to produce an output (the value vector). Writing to memory means updating the parameters of the neural network to encode desired information. We hypothesize that modelling memory as a neural function will offer compression and generalization beyond that of soft look-up tables: deep neural networks are powerful function approximators capable of both strong generalization and memorization [50, 12, 11], and their space overhead is constant.

For a neural network to operate as a useful memory module, it must be possible to record memories rapidly, i.e., to update the network in one shot based on a single datum. We address this challenge by borrowing techniques from few-shot learning through *metalearning* [1, 33, 6, 27]. Recent progress in this domain has shown how models can learn to implement data-efficient, gradient-descent-like update procedures that optimize their own parameters. To store information rapidly in memory, we propose a novel, layer-wise learned update rule. This update modifies the memory parameters to minimize the difference between the neural function's predicted output (in response to a key) and a target value. We find our novel update rule to offer faster convergence than gradient-based update rules used commonly in the metalearning literature [7].

We combine our proposed neural memory module with an RNN controller and train the full model end-to-end (Figure 1). Our model *learns to remember*: we meta-train its reading and writing mechanisms to store information rapidly and reliably. Meta-training also promotes incremental storage, as discussed in §3.5. We demonstrate the effectiveness of our metalearned neural memory (MNM) on a diverse set of learning problems, which includes several algorithmic tasks, synthetic question answering on the bAbI dataset, and maze exploration via reinforcement learning. Our model achieves strong performance on all of these benchmarks.

## 2 Related Work

Several neural architectures have been proposed recently that combine a controller and a memory module. Neural Turing Machines (NTMs) extend recurrent neural networks (RNNs) with an external memory matrix [8]. The RNN controller interacts with this matrix using read and write heads. Despite NTM's sophisticated architecture, it is unstable and difficult to train. A possible explanation is that its fixed-size matrix and the lack of a deallocation mechanism lead to information collisions in memory. Neural Semantic Encoders [28] address this drawback by way of a variable-size memory matrix and by writing new content to the most recently read memory entry. The Differentiable Neural Computer [9] maintains memory usage statistics to prevent information collision while still relying on a fixed-size memory matrix. Memory Networks [46, 40] circumvent capacity issues with a read-only memory matrix that scales with the number of inputs. The read-out functions of all these models and related variants (e.g., [20]) are based on a differentiable attention step [3] that takes a convex combination of all stored memory vectors. Unfortunately, this attention step does not scale well to large memory arrays [32].

Another line of work incorporates dynamic, so-called "fast" weights [14, 38, 37] into the recurrent connections of RNNs to serve as writable memory. For instance, the Hebbian learning rule [13] has been explored extensively in learning fast-weight matrices for memory [38, 2, 23, 26]. HyperNetworks generate context dependent weights as dynamic scaling terms for the weights of an RNN [10] and are closely related to conditional normalization techniques [21, 5, 31].

Gradient-based fast weights have also been studied in the context of metalearning. Meta Networks [27] define fast and slow weight branches in a single layer and train a meta-learner that generates fast weights, while conditionally shifted neurons [29] map loss gradients to fast biases, in both cases for one-shot adaptation of a classification network. Our proposed memory controller adapts its neural memory model through a set of input and output pairs (called interaction vectors below) without directly interacting with the memory weights. Another related approach from the metalearning literature [16, 43, 35, 33, 24] is MAML [6]. MAML discovers a parameter initialization from which a few steps of gradient descent rapidly adapt a model to several related tasks.

Recently, [47, 48] extended the Sparse Distributed Memory (SDM) of [18] as a generative memory mechanism [49], wherein the content matrix is parameterized as a linear Gaussian model. Memory access then corresponds to an iterative inference procedure. Memory mechanisms based on iterative

and/or neural functions, as in [47, 48] and this work, are also related to frameworks that cast memory as dynamical systems of attractors (for some background, see [42]).

# 3 Proposed Method

At a high level, our proposed memory-augmented model operates as follows. At each time step, the controller RNN receives an external input. Based on this input and its internal state, the controller produces a set of memory *interaction vectors*.

In the process of *reading*, the controller passes a subset of these vectors, the read-in keys, to the neural memory function. The memory function outputs a read-out value (i.e., a memory recall) in response. In the process of *writing*, the controller updates the memory function based on the remaining subset of interaction vectors: the write-in keys and target values.

We investigate two ways to bind keys to values in the neural memory: (i) by applying one step of modulated gradient descent to the memory function's parameters (§3.3); or (ii) by applying a learned local update rule to the parameters (§3.4). The parameter update reduces the error between the memory function's predicted output in response to the write-in key and the target value. It may be used to create a new association or strengthen existing associations in memory.

Finally, based on its internal state and the memory read-out, the controller produces an output vector for use in some downstream task. We meta-train the controller and the neural memory end-to-end to learn effective memory access procedures (§3.5), and call the proposed model the *Metalearned Neural Memory* (MNM). In the sections below we describe its components in detail. Figure 1 illustrates the MNM model schematically.

## 3.1 The Controller

The controller is a function $g_\theta$ with parameters $\theta = \{W, b\}$. It uses the LSTM architecture [15] as its core. At each time step it takes in the current external input, $x_t \in \mathbb{R}^{d_i}$, along with the previous memory read-out value $v_{t-1}^r \in \mathbb{R}^{d_v}$ and hidden state $h_{t-1} \in \mathbb{R}^{d_h}$. It outputs a new hidden state: $h_t = \text{LSTM}(x_t, v_{t-1}^r, h_{t-1})$. The controller also produces an output vector to pass to external modules (e.g., a classification layer) for use in a downstream task. The output is computed as $y_t = W_y[h_t; v_t^r] + b_y \in \mathbb{R}^{d_o}$ and depends on the memory read-out vector $v_t^r$. The read-out vector is computed by the memory function, as described in §3.2.

From the controller's hidden state $h_t$ we obtain a set of interaction vectors for reading from and writing to the memory function. These include read-in keys $k_t^r \in \mathbb{R}^{d_k}$, write-in keys $k_t^w \in \mathbb{R}^{d_k}$, target values $v_t^w \in \mathbb{R}^{d_v}$, and a rate vector $\beta_t' \in \mathbb{R}^{d_k}$:

$$[k_{t,1}^r; \dots ; k_{t,H}^r; k_{t,1}^w; \dots ; k_{t,H}^w; v_{t,1}^w \dots ; v_{t,H}^w; \beta_t'] = \tanh(W_v h_t + b_v). \tag{1}$$

The controller outputs $H$ vectors of each interaction type, where $H$ is the number of parallel interaction heads. The single rate vector is further projected down to a scalar and squashed into $[0, 1]$: $\beta_t = \text{sigmoid}(W_\beta \beta_t' + b_\beta)$. Rate $\beta_t$ controls the strength with which the corresponding (key, value) pairs should be stored in memory. The write-in keys and target values determine the content to be stored, whereas the read-in keys are used to retrieve content from the memory. We use separate keys and values for reading and writing because the model interacts with its memory in two distinct modes at each time step: It reads information stored there at previous time steps that it deems useful to the task at hand, and it writes information related to the current input that it deems will be useful in the future. The rate $\beta_t$ enables the controller to influence the dynamics of the gradient-based and local update procedures that encode information in memory (§3.3 and §3.4).

## 3.2 The Memory Function

We model external memory as an adaptive function, $f_{\phi_t}$, parameterized as a feed-forward neural network with weights $\phi_t = \{M^l\}$. Note that these weights, unlike the controller parameters $\theta$, change rapidly from time step to time step and store associative bindings as the model encodes information. Reading from the memory corresponds to feeding the set of read-in keys through the memory function to generate a set of read-out values, $\{v_{t,i}^r\} = f_{\phi_t}(\{k_{t,i}^r\})$.

At hidden layer $l$, the memory function's forward computation is defined as $z^l = \sigma(M^l z^{l-1})$, where $\sigma$ is a nonlinear activation function, $z^l \in \mathbb{R}^{D_l}$ is the layer's activation vector, and we have dropped the time-step index. We execute this computation in parallel on the set of $H$ read-in keys $k^r_{t,i}$ at each time step, yielding $H$ read-out value vectors. We take their mean to construct the single read-out value $v^r_t$ that feeds back into the controller and the output computation for $y_t$.

## 3.3 Writing to Memory with Gradient Descent

A write to the memory consists in rapidly binding the write-in keys $\{k^w_{t,i}\}$ to map to the target values $\{v^w_{t,i}\}$ in the parameters of the neural memory. One way to do this is by updating the memory parameters to optimize an objective that encourages binding. We denote this memory objective by $\mathcal{L}^{\text{up}}_t$ and in this work implement it as a simple mean-squared error:

$$\mathcal{L}^{\text{up}}_t = \frac{1}{H} \sum_{i=1}^{H} ||f_{\phi_{t-1}}(k^w_{t,i}) - v^w_{t,i}||^2_2 \tag{2}$$

We aim to encode the target values $\{v^w_{t,i}\}$ obtained from the controller by optimizing Eq. 2. We obtain a set of memory prediction values, $\{\hat{v}^w_{t,i}\} = f_{\phi_{t-1}}(\{k^w_{t,i}\})$, by feeding the controller's write-in keys through the memory function as parameterized at the previous time step (by $\phi_{t-1}$). The model binds the write-in keys to the target values at time $t$ by taking a gradient step to minimize $\mathcal{L}^{\text{up}}_t$:

$$\phi_t \leftarrow \phi_{t-1} - \beta_t \nabla_{\phi_{t-1}} \mathcal{L}^{\text{up}}_t. \tag{3}$$

Here, $\beta_t$ is the update rate obtained from the controller, which modulates the size of the gradient step. In principle, by diminishing the update rate, the controller can effectively avoid writing into memory, achieving an effect similar to a gated memory update [8].

In experiments we find that the mean-squared error is an effective memory objective: minimizing it encodes the target values in the memory function, in the sense that they can be read out (approximately) by passing in the corresponding keys.

## 3.4 Writing to Memory with a Learned Local Update

Writing to memory with gradient descent poses challenges. Writing a new item to a look-up table can be as simple as adding an element to the corresponding array. In a neural memory, by contrast, multiple costly gradient steps may be required to store a key-value vector pair reliably in the parameters. Memory parameter updates are expensive because, in end-to-end training, they require computation of higher-order gradients (see §3.5; this issue is common in gradient-based metalearning). Sequential back-propagation of the memory error through the layers of the memory model also adds a computational bottleneck.

Other researchers have recognized this problem and proposed possible solutions. For example, the direct feedback alignment algorithm [30], a variant of the feedback alignment method [22], enables weight updates via error alignment with random skip connections. Because these feedback connections are not computed nor used sequentially, updates can be parallelized for speed. However, fixed random feedback connections may be inefficient. The synthetic gradient methods [17] train a model to locally predict the error gradient and this requires the true gradient as a target.

We propose a memory writing mechanism that is fast and gradient-free. The key idea is to represent each neural memory layer with decoupled forward computation and backward feedback prediction functions (BFPF) and perform local updates to the memory layers. Unlike feedback alignment methods, the BFPF and the local update rules are then meta-trained, jointly. Concretely, for neural memory layer $l$, the BFPF is a function $q^l_\psi$ with trainable parameter $\psi_l$ that makes a prediction for an expected activation as: $z'^l = q^l_\psi(v^w_t)$. We then adopt the perceptron learning rule [34] to update the layer locally:

$$M^l_t \leftarrow M^l_{t-1} - \beta^l_t (z^l - z'^l) z^{l-1^T} \tag{4}$$

where $\beta^l_t$ is the local update rate that can be learned for each layer with the controller or separately with the BFPF $q^l_\psi$. The perceptron update rule uses the predicted activation as a true target and approximate the loss gradient w.r.t $M^l_{t-1}$ via $\beta^l_t(z^l - z'^l)z^{l-1^T}$. Therefore, the (approximate) gradient

is near zero when $z^l \approx z'^l$ and there are no changes to the weights. But if the predicted and the forward activations don't match, we update the weights such that the predicted activations can be reconstructed from the weights given the activations $z^{l-1}$ from the previous layer $l-1$.

Therefore, the BFPF module first proposes a regression problem locally for each layer and then the perceptron learning mechanism here solves the local regression problem. One can use a different local update mechanism, rather than the perceptron method. Note that it is helpful to choose the update mechanism that is differentiable w.r.t to its solution to the problem, since the BFPF module is trained to propose problems whose solutions minimize the meta and task loss (§3.5).

With the proposed local and gradient-free update rule, the neural memory writes to its weights in parallel and its computation graph need not be tracked during writing. This makes it straightforward to add complex structural biases, such as recurrence, into the neural memory itself. The proposed approach can readily be applied in the few-shot learning setup as well. For example, we can utilize the learned local update method as an inner-loop adaptation mechanism in a model agnostic way. We leave this to future work.

### 3.5 End-to-end Training via Meta and Task Objectives

It is important to note that the memory objective function, Eq. 2, and the memory updates, Eqs. 3, 4, modify the neural memory function; these updates occur even at test time, as the model processes and records information (they require no external labels). We train the controller parameters $\theta$ and the memory initialization $\phi_0$ and the BFPF parameters $\psi$ end-to-end through the combination of a task objective and a meta objective. The former, denoted by $\mathcal{L}^{\text{task}}$, is specific to the task the model performs, e.g., classification or regression. The meta objective, $\mathcal{L}^{\text{meta}}$, encourages the controller to learn effective update strategies for the neural memory that work well across tasks. These two objectives take account of the model's behavior over an episode, i.e., a sequence of time steps, $t = 1, \ldots, T$, in which the model performs some temporally extended task (like question answering or maze navigation).

For the meta objective, we make use again of the mean squared error between the memory prediction values and the target values, as in Eq. 2. For the meta objective, however, we obtain the prediction values from the *updated* neural function at each time step, $f_{\phi_t}$, after the update step in Eq. 3 or Eq. 4 has been applied. We also introduce a recall delay parameter, $\tau$:

$$\mathcal{L}^{\text{meta}} = \frac{1}{TH} \sum_{\tau=0}^{\mathcal{T}} \sum_{t=1}^{T} \sum_{i=1}^{H} \lambda_\tau ||f_{\phi_t}(k_{t-\tau,i}^w) - v_{t-\tau,i}^w||_2^2 \tag{5}$$

The sum over $\tau$ can be used to reinforce older memory values, and $\lambda_\tau$ is a decay weight for the importance of older (key, value) pairs. The latter can be used, for instance, to implement a kind of exponential decay. We found that $\lambda_\tau$ is task specific; in this work, we always set the maximum recall delay as $\mathcal{T} = 0$ to focus on reliable storage of new information.

At the end of each episode, we take gradients of the meta objective and the task objective with respect to the controller parameters, $\theta$, and use these for training the controller:

$$\theta \leftarrow \theta - \nabla_\theta \mathcal{L}^{\text{task}} - \nabla_\theta \mathcal{L}^{\text{meta}}. \tag{6}$$

These gradients propagate back through the memory-function updates (requiring higher-order gradients if using Eq. 3) so that the controller learns *how* to modify the memory parameters via the interaction vectors (Eq. 1) and the gradient steps or local updates (Eq. 3 or Eq. 4, respectively).

We attempted to learn the memory initialization (similar to MAML) by updating the initial parameters $\phi_0$ w.r.t. the meta loss. We found that this led to severe overfitting. Therefore, we initialize the memory function *tabula rasa* at each task episode from a fixed random parameter set.

Optimizing the episodic task objective will often require the memory to recall information stored many time steps back, after newer information has also been written. This requirement, along with the fact that the optimization involves propagating gradients back through the memory update steps, promotes incremental learning in the memory function, because overwriting previously stored information would harm task performance.

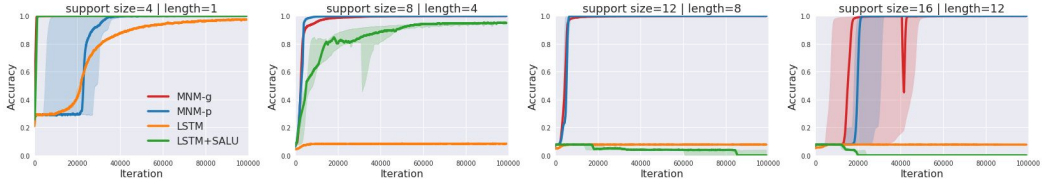

Figure 2: Training curves on the dictionary inference task.

# 4 Experimental Evaluation and Analysis

## 4.1 Algorithmic Tasks

We first introduce a synthetic dictionary inference task to test MNM's capacity to store and recall associated information. This can be considered a toy translation problem. To construct it, we randomly partition the 26 letters of the English alphabet evenly into a source and a target vocabulary, and define a random, bijective mapping $F$ between the two sets. Following the few-shot learning setup, we then construct a support set of $k$ source sequences with their corresponding target sequences. Each source sequence consists of $l$ letters randomly sampled (with replacement) from the source vocabulary, which are then mapped to the corresponding target sequence using $F$. The objective of the task is to predict the target given a previously unseen source sequence whose composing letters have been observed in the support set. For example, after observing the support examples abc→def;tla→qzd, the model is expected to translate input sequence tca to the output qfd.

The difficulty of the task varies depending on the sequence length $l$ and the size of the support set $k$. Longer sequences introduce long-term dependencies, whereas a larger number of observations requires efficient memory structure and compression. We constructed four different task instances with support set size of 4, 8, 12, and 16 and sequence length of 1, 4, 8 and 12.

We trained the MNM models with both gradient-based (MNM-g) and local memory updates (MNM-p). The models have a controller network with 100 hidden units and a three-layer feed-forward memory with 100 hidden units and $\tanh$ activation. We compare against two baseline models: a vanilla LSTM model and a memory-augmented model with the soft-attention look-up table as memory (LSTM+SALU). In the LSTM+SALU model, we replace the feed-forward neural memory with the look-up table, providing an unbounded memory to the LSTM controller. The training setup is given in Appendix A.

Figure 2 shows the results for our dictionary inference task (averaged over 5 runs). All models solved the first task instance and all memory-augmented models converged for the second case. As the task difficulty increased for the last two cases, only the MNM models converged and solved the task with zero error.

In Figure 3, we compared the training wallclock time and the memory size of MNM(-g, -p) against LSTM+SALU models on these task runs. When the input length is small, the LSTM+SALU model is faster than MNM-g and similar in speed to MNM-p, and has a smaller memory footprint than both. However, as soon as the input length exceeds the size of the MNM memory's hidden units, LSTM+SALU becomes less efficient. It exhibits quadratic growth whereas the MNM models grow approximately linearly in the wallclock time. Figure 3's left plot also demonstrates that that learned

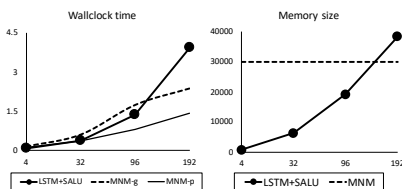

Figure 3: Model comparison over varying input lengths ($x$-axes).

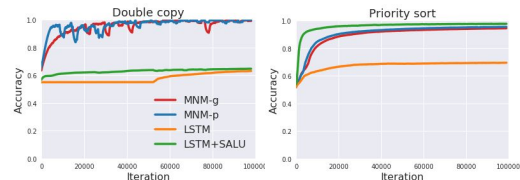

Figure 4: Training curves on programming tasks.

Table 1: Results on bAbI question answering.

|  | Sentence-level | | Word-level | | | |
|---|---|---|---|---|---|---|
|  | EntNet | TPR-RNN | DNC | SDNC | MNM-g | MNM-p |
| Mean Error | $9.7 \pm 2.6$ | $1.34 \pm 0.52$ | $12.8 \pm 4.7$ | $6.4 \pm 2.5$ | $3.2 \pm 0.5$ | **0.55 ±0.74** |
| Failed Tasks ($> 5\%$ error) | $5 \pm 1.2$ | $0.86 \pm 1.11$ | $8.2 \pm 2.5$ | $4.1 \pm 1.6$ | $1.3 \pm 0.8$ | **0.08 ± 0.28** |

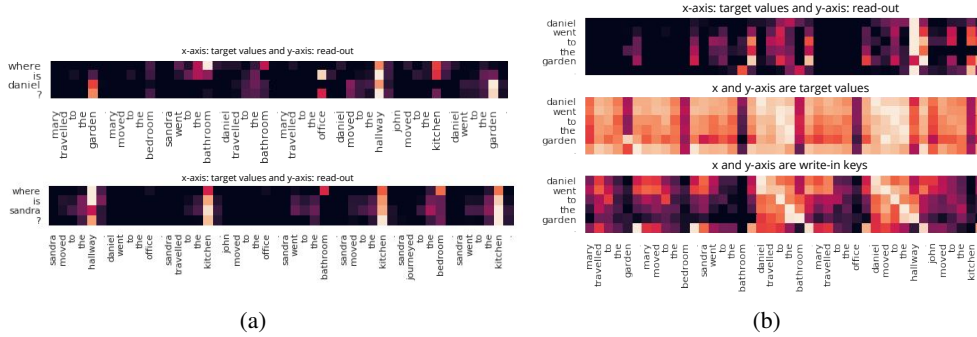

(a)                                          (b)

Figure 5: Visualization of learned memory operations. (a) At each question word (y-axis), the model recalls memory contents written for entities in the story (x-axis) that are most closely related to the question type (e.g., locations for *where* questions). (b) Towards the end of a story (y-axis), the model learns to access and update the memory conditioned on structured information (e.g., location and character) memorized earlier in the story (x-axis).

local updates (MNM-p) confer significant speed benefits over gradient-based updates (MNM-g), since the former can be applied in parallel.

We further evaluated these models on two standard memorization tasks: double copy and priority sort. As shown in Figure 4, the MNM models quickly solve the double copy task with input length 50. On the priority sort problem, the LSTM+SALU model demonstrated the strongest result. This is surprising, since the task was previously shown to be hard to solve [8]. It suggests that the unbounded memory table and the look-up operation are especially useful for sorting. The MNM models' strong performance across the suite of algorithmic tasks, which require precise recall of past information, indicates that MNM can store information reliably.

## 4.2    bAbI Question Answering

bAbI is a synthetic question-answering benchmark that has been widely adopted to evaluate long-term memory and reasoning [45]. It presents 20 reasoning tasks in total. We aim to solve all of them with one generic model. Previous memory-augmented models addressed the problem with sentence-level [40, 20] or word-level inputs [9, 32]. Solving the task based on word-level input is harder, but more general [36]. We trained word-level MNM models following the setup for the DNC [9].

The results are summarized in Table 1. The MNM model with the learned local update solved all the bAbI tasks with near zero error, outperforming the result of TPR-RNN [36] (previously the best model operating at sentence-level). It also outperformed the DNC by around 12% and the Sparse DNC [9] by around 6% in terms of mean error. We report the best results and all 12 runs of MNM-p in Appendix B. MNM-g with the gradient-based update solved 19 tasks, failing to solve only the basic induction task.

We also attempted to train a word-level LSTM+SALU baseline as described in the previous section. However, multiple LSTM+SALU runs did not solve any task and converged to 77.5% mean error after 62K training iterations. With the same number of iterations, the MNM runs converged to a 9.5% mean error and solved 16.5 tasks on average. This suggests the importance of a deep neural memory and a learned memory access mechanism for reasoning.

**Analyzing Learned Memory Operations:** To understand the MNM more deeply, we analyzed its temporal dynamics through the similarity between the keys and values of its read/write operations as it processed bAbI. Here cosine distance is used as a similarity metric. We found that the neural memory exhibits readily interpretable structures as well as efficient self-organization.

Intuitively, keys and values correspond to the locations and contents of memory read/write operations. Consequently the temporal comparison of various combinations of these vectors can have meaningful interpretations. For example, given two time steps $t_1 < t_2$, when comparing $v_{t_2}^r$ against $v_{t_1}^w$, higher similarity (brighter colors in Figure 5) indicates that the content stored at $t_1$ is being retrieved at $t_2$. We first applied this comparison to a bAbI story (x-axis in Figure 5a) and the corresponding question (y-axis), since they are read consecutively by the model. Upon reading the question word "where", the model successfully retrieves the location-related memories. When the model reads in the character names, the retrieval is then "filtered" down to only the locations of the characters in question. Furthermore, the retrieval also appears to be closer to more recent locations, effectively modeling this strong prior in the data distribution of bAbI.

Similarly, we analyzed the memory operations towards the end of a story (y-axis) and examined how the model uses the memory developed earlier (x-axis). Again, comparing $v_{t_2}^r$ and $v_{t_1}^w$ (row 1 in Figure 5b), the bright vertical stripe at "hallway" indicates that the memory retrieval focuses more on Daniel's most recent location (while ignoring both his previous locations and locations of other characters). In addition, $v_{t_2}^w$ and $v_{t_1}^w$ are compared in row 2, Figure 5b, where the dark vertical stripes indicate that the memory is being updated aggressively with new contents whenever a new location is mentioned — potentially establishing new associations between locations and characters. In the comparison between $k_{t_2}^w$ and $k_{t_1}^w$ (row 3 in Figure 5b), two bright diagonals are observed in the sentences related to the matching character Daniel, suggesting that (a) the model has likely acquired an entity-based structure and (b) it is capable of leveraging this structure for efficient memory reuse.

More examples can be found in the appendix. Overall, the patterns above are salient and consistent, indicating our model's ability to disentangle objects and their roles from a story, and to use that information to dynamically establish and update associations in a structured, efficient manner — all of which are key to neural-symbolic reasoning [39] and effective generalization.

## 4.3 Maze Exploration by Reinforcement Learning

External memory may be helpful or even necessary for agents operating in partially observable environments, where current decision making depends on a sequence of past observations. However, memory mechanisms also add complexity to an agent's learning process [19], since the agent must learn to access its memory during experience. In our final set of experiments, we train an RL agent augmented with our metalearned neural memory. We wish to discover whether an MNM agent can perform well in a sequential decision making problem and use its memory to improve performance or sample efficiency.

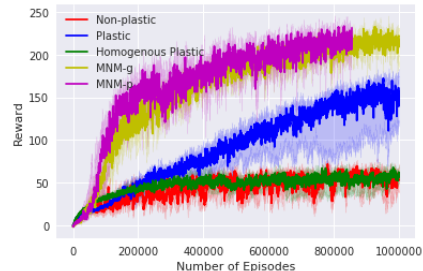

Figure 6: Training curves on the maze exploration task.

We train MNM agents on a maze exploration task from the literature on meta-reinforcement learning [4, 44]. Specifically, we adopted the grid world setup from [23]. In this task, for each episode, the agent explores a grid-based maze and attempts to reach a goal position. Reaching the goal earns the agent a reward of 10 and relocates it to a random position. The agent must then return to the goal to collect more reward before the episode terminates. To the agent, the goal is invisible in the maze and its position is chosen randomly at the beginning of each episode. Inputs to the agent are its surrounding $3 \times 3$ cells, the time step, and the reward from the previous time step. The agent receives -0.1 reward for hitting a wall and 0 reward otherwise. A $9 \times 9$ grid maze is shown in Figure 7 (Appendix A) for illustration.

We trained agents on a $9 \times 9$ maze following the setup of [23] to provide a direct comparison with the differential plasticity agents of that work. We used the Advantage Actor-Critic (A2C) algorithm for optimization, a non-asynchronous variant of the A3C method [25]. The MNM agent has a neural memory and controller with 100 and 200 hidden units, respectively. The training curve for the $9 \times 9$ maze (averaged over 10 runs) is plotted in Figure 6, along with results from [23]. As can be seen, the agents with *differential plasticity* (denoted Plastic and Homogenous Plastic) converge to a reward of 175 after training on nearly 1M episodes. MNM, on the other hand, reaches the same reward in only 250K episodes. It obtains significantly higher final reward after 1M episodes. This result shows

that the MNM fosters improved performance and sample efficiency in a sequential decision making scenario and, promisingly, it can be trained in conjunction with an RL policy.

## 5    Conclusion

We cast external memory for neural models as a rapidly adaptable function, itself parameterized as a deep neural network. Our goal was for this memory mechanism to confer the benefits of deep neural networks' expressiveness, generalization, and constant space overhead. In order to write to a neural network memory rapidly, in one shot, and incrementally, such that newly stored information does not distort existing information, we adopted training and algorithmic techniques from metalearning. The proposed memory-augmented model, MNM, was shown to achieve strong performance on a wide variety of learning problems, from supervised question answering to reinforcement learning. Our learned local update algorithm can be applied in an other setup than the memory one. In future work, we will investigate different neural architectures for metalearned memory and the effects of recall delays in the meta objective.

## Acknowledgements

We thank Thomas Miconi for sharing data. We thank Geoff Gordon for helpful comments and suggestions.

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
