[Supplementary Material · MNM_camera_ready_supp.pdf]

Table 2: Hyperparameters used in the experiments

| | Controller | | | | Memory | | | | |
|---|---|---|---|---|---|---|---|---|---|
| | Layers | $d_h$ | $d_o$ | $d_i$ | Layers | $d_k$ | $d_v$ | $D_l$ | $H$ |
| Algorithmic tasks | 1 | 100 | 100 | 100 | 3 | 100 | 100 | 100 | 1 |
| bAbI task | 1 | 256 | 256 | 256 | 3 | 100 | 100 | 100 | 3 |
| Maze expl. | 1 | 200 | 200 | - | 3 | 100 | 100 | 100 | 3 |

## A  Training Details

We used tanh activation function and three layer feed-forward neural net for our memory. For the BFPF $q_\psi^l$, we performed an initial experiment evaluating feed-forward and LSTM architectures with different inputs and a simple single-layer MLP with input $v_t^w$ worked well. The number of parallel heads $H$ was 1, 3 and 3 for the algorithmic, bAbI and maze exploration experiments, respectively.

For dictionary inference task, we used four special input characters in addition to the main vocabulary for the end of a sequence, a support example separator, the end of a support set and an input place holder for target. For double copy task, input sequences of length 50 were constructed by randomly sampling (with replacement) from 10 unique characters. For the sort task, input sequences were length of 20 and consisted of 8-bit binary vectors along with their scalar weights. The model were trained to predict the first 16 vectors sorted. This follows the setup of NTM [8].

For bAbI question answering, we appended each question to the end of its related story and inserted additional placeholders for answer tokens. The models read the story first and then the question word-by-word, and once reaching the answer placeholders produce a prediction. The standard data splits were used in the experiment. We perform an early-stopping based on the standard development set and evaluate on the test set.

The batch sizes were set to 32 and 128 for the algorithmic and bAbI experiments, respectively. All models were optimized using Adam optimizer. The hyperparameters for Adam optimizer were set to default values (alpha=0.001 and beta=0.9) for all learning problems except the RL one. For the RL task, we used the same hyperparameters as [23]. Table 2 lists our model hyperparameters. In Figure 7, we have shown an instance of $9 \times 9$ maze.

## B  Detailed Results on bAbI Task

Table 3 and 4 show the best results of the compared models and the detailed runs of our best performing model.

Table 3: Best results on bAbI question answering.

| Task | Sentence-level | | Word-level | | | |
|---|---|---|---|---|---|---|
| | EntNet | TPR-RNN | DNC | SDNC | MNM-g | MNM-p |
| 1: one supporting fact | 0.1 | 0 | 0 | 0 | 0 | 0 |
| 2: two supporting facts | 2.8 | 0.4 | 0.4 | 0.6 | 0.2 | 0.1 |
| 3: three supporting facts | 10.6 | 3.4 | 1.8 | 0.7 | 1.8 | 0.9 |
| 4: two argument rel. | 0 | 0.2 | 0 | 0 | 0 | 0 |
| 5: three argument rel. | 0.4 | 1.0 | 0.8 | 0.3 | 0.4 | 0.3 |
| 6: yes/no questions | 0.3 | 0.1 | 0 | 0 | 0 | 0 |
| 7: counting | 0.8 | 1.0 | 0.6 | 0.2 | 0.2 | 0.3 |
| 8: lists/sets | 0.1 | 0.5 | 0.3 | 0.2 | 0.2 | 0 |
| 9: simple negation | 0 | 0.3 | 0.2 | 0 | 0 | 0 |
| 10: indefinite kd. | 0 | 0.4 | 0.2 | 0.2 | 0.1 | 0 |
| 11: basic coref. | 0 | 1.3 | 0 | 0 | 0 | 0 |
| 12: conjunction | 0 | 0.2 | 0 | 0.1 | 0 | 0 |
| 13: compound coref. | 0 | 2.1 | 0 | 0.1 | 0 | 0 |
| 14: time reasoning | 3.6 | 0.2 | 0.4 | 0.1 | 0.5 | 0.1 |
| 15: basic deduction | 0 | 0 | 0 | 0 | 0 | 0 |
| 16: basic induction | 52.1 | 0.4 | 55.1 | 54.1 | 51.2 | 0.7 |
| 17: positional reasoning | 11.7 | 0.6 | 12.0 | 0.3 | 0 | 0 |
| 18: size reasoning | 2.1 | 0 | 0.8 | 0.1 | 0 | 0.2 |
| 19: path finding | 63.0 | 4.2 | 3.9 | 1.2 | 0.7 | 0.9 |
| 20: agent's motivation | 0 | 0 | 0 | 0 | 0 | 0 |
| Mean Error: | 7.38 | 0.81 | 3.8 | 2.9 | 2.76 | **0.175** |
| Failed Tasks (> 5% error): | 4 | 0 | 2 | 1 | 1 | **0** |

Table 4: Results from 12 runs of MNM-p model.

| Task | run-1 | run-2 | run-3 | run-4 | run-5 | run-6 | run-7 | run-8 | run-9 | run-10 | run-11 | run-12 | Mean | Best |
|---|---|---|---|---|---|---|---|---|---|---|---|---|---|---|
| 1: one supporting fact | 0 | 0 | 0 | 0 | 0 | 0 | 0 | 0 | 0 | 0 | 0 | 0 | $0 \pm 0$ | 0 |
| 2: two supporting facts | 0.2 | 0 | 0.7 | 0.1 | 0.1 | 0.3 | 0.4 | 0 | 0.2 | 0.1 | 0.3 | 0 | $0.2 \pm 0.2$ | 0 |
| 3: three supporting facts | 2 | 2.1 | 2.2 | 1.7 | 0.9 | 1.1 | 1.9 | 1.5 | 2 | 2.1 | 2.4 | 1.7 | $1.8 \pm 0.43$ | 0.9 |
| 4: two argument rel. | 0 | 0 | 0 | 0 | 0 | 0 | 0 | 0 | 0 | 0 | 0 | 0 | $0 \pm 0$ | 0 |
| 5: three argument rel. | 0.5 | 0.8 | 1.1 | 0.4 | 0.3 | 0.7 | 0.2 | 0.8 | 0.8 | 0.4 | 0.4 | 0.7 | $0.59 \pm 0.25$ | 0.2 |
| 6: yes/no questions | 0 | 0.2 | 0 | 0 | 0 | 0 | 0 | 0 | 0 | 0.1 | 0.1 | 0 | $0.03 \pm 0.06$ | 0 |
| 7: counting | 0 | 0 | 0 | 0 | 0.3 | 0.2 | 0.3 | 0.3 | 0.1 | 0.4 | 0 | 0 | $0.13 \pm 0.15$ | 0 |
| 8: lists/sets | 0 | 0 | 0 | 0.1 | 0 | 0.1 | 0 | 0 | 0.1 | 0.1 | 0 | 0 | $0.03 \pm 0.05$ | 0 |
| 9: simple negation | 0.1 | 0 | 0 | 0 | 0 | 0 | 0 | 0 | 0 | 0 | 0 | 0 | $0.01 \pm 0.03$ | 0 |
| 10: indefinite kd. | 0 | 0.1 | 0.1 | 0.1 | 0 | 0 | 0 | 0 | 0.1 | 0 | 0 | 0.1 | $0.04 \pm 0.05$ | 0 |
| 11: basic coref. | 0 | 0 | 0 | 0 | 0 | 0 | 0 | 0 | 0 | 0 | 0 | 0 | $0 \pm 0$ | 0 |
| 12: conjunction | 0 | 0 | 0 | 0.1 | 0 | 0 | 0 | 0 | 0 | 0 | 0 | 0 | $0.01 \pm 0.03$ | 0 |
| 13: compound coref. | 0 | 0 | 0 | 0 | 0 | 0 | 0 | 0 | 0 | 0 | 0 | 0 | $0 \pm 0$ | 0 |
| 14: time reasoning | 0.1 | 0.8 | 3.2 | 1.3 | 0.1 | 0.3 | 1.9 | 1 | 3.7 | 2.2 | 3.1 | 0.2 | $1.49 \pm 1.25$ | 0.1 |
| 15: basic deduction | 0 | 0 | 0 | 0 | 0 | 0 | 0 | 0 | 0 | 0 | 0 | 0 | $0 \pm 0$ | 0 |
| 16: basic induction | 0.5 | 0.7 | 49.2 | 0.6 | 0.7 | 1 | 0.9 | 0.3 | 0.4 | 0.3 | 0.4 | 0.5 | $4.63 \pm 13.44$ | 0.3 |
| 17: positional reasoning | 0.5 | 0.3 | 1.9 | 0 | 0 | 0 | 0 | 0 | 0 | 1.5 | 1.3 | 0 | $0.46 \pm 0.67$ | 0 |
| 18: size reasoning | 0.1 | 0.2 | 0.1 | 0.6 | 0.2 | 0.1 | 0.3 | 0.1 | 0.2 | 0 | 0.5 | 0.6 | $0.25 \pm 0.2$ | 0 |
| 19: path finding | 1.4 | 3.1 | 0.7 | 4.2 | 0.9 | 0.2 | 0.6 | 0.1 | 0.1 | 1.4 | 2.1 | 0 | $1.23 \pm 1.26$ | 0 |
| 20: agent's motivation | 0 | 0 | 0 | 0 | 0 | 0 | 0 | 0 | 0 | 0 | 0 | 0 | $0 \pm 0$ | 0 |

Figure 7: An example maze of $9 \times 9$ size. The current goal and agent locations are indicated in yellow and green, respectively.

## C   Relation to Sparse Distributed Memory

Sparse distributed memory [18] can be seen as a two-layer neural network in which the first layer outputs an address to read from or write to and the second layer encodes the memory content. Concretely, given fixed address matrix $A \in \{-1, 1\}^{D_l \times d_v}$ and content matrix $C_t \in \mathbb{Z}^{d_v \times D_l}$, sparse distributed memory first calculates an activation vector $a_t \in$ according to

$$a_t = \sigma(A k_t), \tag{7}$$

where $k_t$ is an input and $\sigma(m)$ is an element-wise function that outputs 1 if $\frac{1}{2}(D_l - m) \leq \delta$ and 0 otherwise, with $\delta$ a threshold. At read time, a memory output $\hat{v}_t$ is obtained by multiplying the activation vector by the content matrix: $\hat{v}_t = C_t a_t$. At write time, the memory is updated to store a content vector $v_t$ according to:

$$C_t = C_{t-1} + v_t a_t^T. \tag{8}$$

We can derive related computations for a memory function parameterized as a two-layer feed-forward neural network ($L = 2$). The computation at the first layer is $a_t = \sigma(M_t^1 k_t)$, which becomes identical to that of the SDM if we use the address matrix $A$ for $M_t^1$ and the binary activation function for $\sigma$. At read time, the computation at the second layer is $\hat{v}_t = M_t^2 a_t$, where $M_t^2$ is analogous to the content matrix $C_t$.

For our memory update, the derivative of the mean squared error with respect to the second-layer parameter matrix is:

$$\frac{\partial \mathcal{L}_t^{\text{up}}}{\partial M_t^2} = \frac{2}{d_v}(\hat{v}_t - v_t)a_t^T. \tag{9}$$

Then we can rewrite the update for the last layer as

$$M_t^2 = M_{t-1}^2 + v_t a_t^T - \hat{v}_t a_t^T, \tag{10}$$

where we have dropped the normalization constant $\frac{2}{d_v}$ and set the update rate $\beta_t$ to one for convenience. Comparing Eq. 10 with Eq. 8, the gradient-based update incorporates the SDM update rule as a component and performs a slightly smarter computation, since it makes no update whenever the desired value is already stored (i.e., $\hat{v}_t = v_t$).

## D   Visualization of Learned Memory Operation