[Reviews · NeurIPS 2019]

Reviewer 1



UPDATED I think the authors for their rebuttal comments. All my concerns have been addressed (modulo seeing the extra results / error bars) so I am raising my score to 8. The idea of parameterising the memory as a neural network, and using ideas from metalearning to quickly train it to produce a specified output for new sequences, is very interesting and novel. The paper is overall well written, and I believe should be reproducable by those familiar with metalearning approaches. The justification for the model is interesting - essentially instead of writing some values to a fixed size memory, and then reads being limited to a convex combination of the written values, using a neural network allows potential benefits with compression, as well as generalisation, with constant space. Obviously the key issue with this is whether the memory function can be easily modified in one shot so that a new set of keys and values will be 'read' approximately correctly. The method description is clear, and many parts will feel very familiar to those experienced with NTM / DNC. The memory function appears to be a 'standard' feed forward MLP, which is run on a batch of read-in keys generated by the RNN controller at each timestep. The bulk of the method deals with how to get the memory function to produce the write-out values when it is at some point queried with the write-in keys, which could be at any point including the very next timestep. The Gradient descent approach does an internal step of gradient descent during a forwards pass (including at test time) using a learning rate $\beta_t$ which is also outputted by the controller. This allows the model to disable writing for any given timestep. A meta objective is introduced which encourages the result of this gradient descent step to then produce the correct values when read. For this approach, higher order gradients must be computed so the meta objective can be trained, along with the task-specific objective. I found the fact that this could work well at all, with just a single step, to be quite interesting and surprising. An alternate to this gradient-descent based writing is proposed in terms of a local learning rule. Direct Feedback Alignment is cited as a close inspiration. I feel that [1] would also be a worthwhile citation here, as it is in some sense a generalisation of [Direct]Feedback Alignment, and other methods. Essentially, a parameterised 'backward function' is learnt for each internal layer of the memory function, which produces some error feedback $z'^l$ based on the write-out values which the memory network is supposed to produce. The perceptron learning rule is then deployed to adjust the memory function weights. The same meta-objective is used to ensure that the written values become available, which backpropagates into the weights of these backward functions. Some aspects of this approach were slightly unclear to me - the backwards function is written as taking in $v_t^w$, ie the entire set of write out values at time t. Given that there are in general $H$ of these values, how does the backward function process each of these? Presumably the function should not depend on the ordering of the various write values, so how is this permutation invariance implemented? I can imagine that the backward function operats on a minibatch and then sums / maxes over the batch dimension of outputs, which is reasonable, but I feel not obvious from the text. Another aspect is whether the backwards function could be parameterised on the write-in keys as well - presumably this is not necessary, and in a sense the information from the write-in keys is available in some form inside the memory function activations z_l. However, I feel some brief discussion of this would benefit the text. The authors denote the gradient descent based model as MNM-g and the local updated rule as MNM-p, and evaluate the architectures comparing to reasonable baselines. In the synthetic dictionary task, the authors show that as the sequence length and support size increase, LSTM and LSTM-SALU severely degrade back to random, whereas their methods continue to learn. I think these graphs would benefit hugely from error bars from multiple runs - my experience of training memory models which have the 'delayed takeoff' visible in the rightmost plot in figure 2 is that the takeoff point can vary significantly. Nevertheless, clearly the MNM-* family perform very well here. The bAbI results are excellent, demonstrating a new SOTA. It's interesting that MNM-p wins over MNM-g so significantly - this matches my intuition that writing into a memory function quickly with gradient descent would prove hard (although it clearly works somewhat), and that something smarter (ie the local learning rule) would be needed. The diagrams in figure 5 visualizing memory read and write correlation are very interesting, although I feel are lacking in labelling. The text below does specify what vectors are being compared, presumably something like the cosine distance between the read key/values and the previous write key/values. As far as I can see this is just described as a 'similarity' (L281) which is somewhat ambiguous. More importantly though, to keep track of what pairs of vectors are being compared (various combinations of {read,write} {key,values}) one has to read all the way down until the start of the next page. Simply adding a title to each of the subgraphs in Figure 5 would improve things massively. Notwithstanding this, the graphs do show a variety of interesting behaviours including reads, information that is no longer valid being overwritten, etc. The final experiments on RL are interesting, the maze setup as shown in figure 7 (supplemental) seems very basic to me, hard to call it a maze in any meaningful sense, but the authors are reusing an environment from other work so I am not counting this against them. The results shown MNM-g as outperforming the differential plasticity agents, which is a nice result. However I am totally baffled as to why MNM-p is not included here. Up until now, for me the narrative of the paper is that MNM-p is as good or better than MNM-g, due a somewhat more sophisticated approach, and the fact that it doesn't require higher order gradients presumably means it is somewhat simpler implementation-wise. For the final experiments to be missing what up until now seemed like the best model is very confusing! I would strongly recommend that these curves are added. If it actually performs much worse than MNM-g, then that's fine, as long as some attempt at explanation can be made (eg - "RL gradients are much noisier, and this interferes with our ability to learn a good backward function, therefore MNM-g is better for RL"). MNM-p is SOTA on bAbI, so it's usefulness for at least sometasks is beyond dispute - whether or not it's good for this RL task, the paper still has a strong message that this *pair* of related techniques are valuable additions to the literature. Overall this paper is well written and proposes a very interesting new model. There are a few aspects which could be made clearer, but overall I found this to be a very nice piece of work. [1] - Decoupled neural interfaces using synthetic gradients - Jaderberg et al, ICML 2017

Reviewer 2



This paper introduces a methodology to build and train the neural network that acts like an external memory module through meta-learning. The central idea of building such model is that the memory module has to be updated fast so that it can remembers from writing the contents even for a one shot, therefore they use meta-learning strategy. Q1.Is there any intuition that proposed model has better performance than NTM, which had also trained from episodic training? I wonder if it's better because the model is applied on quite recent meta-training (MAML), or its advanced network modeling. Q2. The experiments and their results are not intuitive to understand. I expect to have the algorithmic tasks conducted in NTM, but they give novel tasks they made even without comparing with previous works. Also, some pictures are too small to read. Are there any comparison for memory-based networks?

Reviewer 3



Update: I read the authors' response as well as the other reviews and found this quite helpful to clarify some points of confusion (such as the task distribution). I would suggest to further clarify this in an updated version. While I was originally unconvinced whether there was sufficient algorithmic novelty on the Meta-Learning side I got convinced that this paper successfully demonstrates how both techniques can be successfully combined which is a valuable message to the community. I'm therefore happy to update my score from 5 to 6. --- - Figure 1: Given that this graphic appears on the first page, I believe that notation should be avoided, especially since none of the values is introduced until section 3. - Section 3: I'm not sure I fully understand the definition of the meta loss in equation (5). In particular, the summations over time steps seems to indicate that we must store H values $v^w_{t-\tau}$ for each time step during training. Is this correct? - It's not clear to me what the exact training set-up is. The authors mention "meta-training" (e.g. Line 108) without a clear definition of the term. Is the model trained on distribution of tasks? What are those distributions for each experiment? - Line 134: Why are biases missing? - Line 137: Notation appears incorrect. The function is not taking a set as an argument but is evaluated for each element in the set. - I was somewhat confused throughout the paper why the authors referred to a function approximator as a memory module. I think most people will think of a memory module as a raw storage of information.

[Author Response · NeurIPS 2019]

We thank the reviewers for their time and helpful feedback. Below we respond to their comments in turn.

**Reviewer 1** 'permutation invariance in write values' Yes, we process the values in parallel and then take the sum over the batch dimension. We will make this clearer in the updated manuscript.

'parameterization of the backwards function' Thanks for this suggestion. We agree this discussion would be useful. In the Appendix, we briefly describe the different parameterization strategies for the backward function that we initially attempted. We'll move this to the main text and add more discussion.

'Error bars for the plots in Figure 2.' We will conduct multiple runs and include error bars in Figure 2.

'Add titles to each subgraph in Figure 5.' Indeed, we should have done this. We will add titles to each subgraph.

'MNM-p results for the Maze exploration task.' At the time of submission, we were still running the MNM-p model on the maze task. The results show that MNM-p performs on par with MNM-g on the maze task. We will include this finding in the paper.

'cite synthetic gradients' Thanks for pointing this out. We'll add that citation.

**Reviewer 2** 'Is there any intuition that proposed model has better performance than NTM ' Our hypothesis is that a memory module implemented as a neural network, whose weights can change over the course of an episode, will offer greater expressivity than attention-based tabular memory modules like the NTM, as well as constant time and space overhead. The main goal for the new dictionary inference task introduced in the paper is to validate this hypothesis empirically. Note that the MNM uses similar network components as the NTM: we use an LSTM controller combined with an MLP memory function (more complex architectures for the memory would be interesting to explore in future work). On the other hand, the meta-training process for writing to memory is indeed based on recent techniques (more on techniques from [1,2] than MAML), as you point out, and our experiments suggest this kind of meta-training can be adopted for information storage over an episode.

'The experiments and their results are not intuitive to understand' The double copy and sort tasks are standard algorithmic benchmarks from the literature on memory-augmented neural networks [3,4]. We apologize if this was not stated clearly and will fix this in the updated manuscript. The LSTM+SALU baseline that we compare against replaces the metalearned MLP memory function with the soft-attention look-up table used in several memory augmented models, eg RNNSearch[5], MemN2N[6] Transformers[7]. Because this is the same overall architecture as MNM with a different memory module, we believe it's a fair, minimally distinct model to compare against to validate the hypothesis stated above. The dictionary inference task we introduced is a toy proxy for few-shot machine translation and we believe it can be used more broadly to study models with adaptive/memory behavior. We will make the small plots larger to improve their readability.

**Reviewer 3** 'Figure 1: notations should be avoided' That makes sense. We will use the name "interaction vectors" in Figure 1 or describe the notations in the caption or move the figure.

'Section 3: the definition of the meta loss' That's correct: during training, we store H values as part of the computation at each time step and use them later for backprop.

'the exact training set-up is unclear' During training, the model sees many instances of a given task. These instances are used for meta-training. For example, in the "sort" algorithmic task it sees many randomly generated sequences to be sorted. The sequence length is 20 and each element in sequence is 8 bits. The model sees 1M randomly generated sequences during training. We plotted the training curve for show convergence. See NTM [3] for a more detailed description. We will clarify the training setup in the updated manuscript.

'Line 134: Why are biases missing?' For simplicity, we didn't use biases in the neural memory module.

'Line 137: Notation appears incorrect' Thanks for pointing this out. We'll fix the notation.

'why the authors referred to a function approximator as a memory module' We cast memory as a family of adaptive functions – neural nets – that are suitable for the storage of information. These functions can be updated rapidly over the course of an episode (not just at training time) to "write in" new information, and they can be read from over an episode to recall information. This is indeed a somewhat novel take on memory, but note that a flexible function approximator could, in theory, learn to approximate more standard "data structure" types of memory functions, like hash maps, etc. Note also that typically memory-augmented neural networks do not store "raw" input information in their memories, but rather vector encodings derived from the inputs.

'the authors could try their method on a harder QA task' Our focus in this work was to validate the model and understand its memory operation by running on bAbI task. Since the results were very encouraging, we plan to put more effort and try the model on other benchmarks in future work.

Ref: [1] Andrychowicz et al. "Learning to learn by gradient descent by gradient descent." Advances in neural information processing systems. 2016. [2] Ravi et al. "Optimization as a model for few-shot learning." (2016). [3] Graves et al. "Neural turing machines." arXiv preprint arXiv:1410.5401 (2014). [4] Santoro et al. " Advances in Neural Information Processing Systems. 2018. [5] Bahdanau et al. "Neural machine translation by jointly learning to align and translate." arXiv preprint arXiv:1409.0473 (2014). [6] Sukhbaatar et al. "End-to-end memory networks." Advances in neural information processing systems. 2015. [7] Vaswani et al. "Attention is all you need." Advances in neural information processing systems. 2017.

[Meta-Review · NeurIPS 2019]

All reviewers give the paper acceptance scores. The author's response was very clear and two of the reviewers have increased their scores. The paper is well written and proposes a novel memory architecture borrowing ideas from recent meta-learning work making it a good contribution to the literature.